Editorial

# Reframing "disappointing" data at *Life Science Alliance*

Tim Fessenden

**DOI** https://doi.org/10.26508/lsa.202603674 | Received 18 February 2026 | Accepted 18 February 2026 | Published online 2 March 2026

## Introduction

What research projects are publication-worthy, and at what stage? Canny researchers develop a sixth sense for "ripe" projects ready for submission to a journal or posting as a preprint. These scientists perceive the moment a collection of results has crossed the threshold into a compelling story, or when a surprising finding has enough support for release to the community. Suddenly, the sum is greater than its parts, and the writing begins.

But is this sixth sense blind to some research projects, numb to results that could be published but are not? Or does it run into the wall of repeated editorial rejections at every journal? One type of research result, negative data, is perennially neglected by both researchers and peer-reviewed journals. Many scientists deem negative results uninteresting. They are not worth the effort to write up, in part because they are not publishable. Researchers who unearth only negative results often feel they must await a turn in their fortunes before there is something to write about. *Life Science Alliance* (LSA) explicitly welcomes submissions describing negative results, and since joining the journal as Executive Editor I've come to appreciate the value of these studies and the larger questions posed by their status.

## What are negative data?

Null or inconclusive results confront us with "nature's apparent silence or nonacquiescence to our expectations," as Stephen Jay Gould put it (Gould, 1993). But thinking of this category as simply any experimental outcome that disproves the researcher's hypothesis, whereas technically true, is immediately apparent as an over-broad definition. Discoveries that reshape entire fields often originate as surprising negative results that contradict a broadly held expectation. Clearly the opposite of disappointing.

Null results are essential to refutations which disprove prior findings and correct the scientific record. Such results may or may not be disappointing to the authors, but their relative scarcity in the published literature is telling. Consider an example: the F-box protein Fbxo7 is linked to risk of Parkinson's disease and was reported to engage Parkin on mitochondria, ultimately triggering mitochondrial clearance via mitophagy (Burchell et al, 2013). Ten years later, Wade Harper and colleagues published results using the same cell culture models to disprove this, finding no interaction between these proteins and no mitophagy defect in cells lacking Fbxo7 (Kraus et al, 2023). A journal editor might have concluded that the study makes an insufficient advance in our understanding of Fbxo7 - telling us only what the protein does not do.

Refutation or not, negative results appear to teach us nothing new about the natural world. They are disappointing in this sense, properly understood only through the human desire for discovery and understanding. "The issue of negative data," remarked ocean ecologist Howard Browman, "reflects our training, our thoughtfulness about what we do as scientists (and how we do it), and our humanity, with all its inherent biases" (Browman, 1999).

These biases manifest differently across disciplines, with effects specific to each field. The medical literature has long struggled with uneven publication of clinical trial results (Turner et al, 2008), while chemists tend not to report failed reaction conditions (Taniike & Takahashi, 2023). In his essay Cordelia's Dilemma, Gould recalls that stasis or non-change throughout the fossil record was discounted as "an embarrassing feature" … "best ignored as a manifestation of nothing (that is, nonevolution)" (Gould, 1993). Stasis was simply not reported in the literature until, of course, his theory with Niles Eldredge of punctuated equilibrium. Spend enough time in any given field and one learns which observations are widely agreed to be uninteresting, unworthy of writing up into a manuscript, and similarly unwanted by journal editors. In this sense, there is not a monolithic bias against negative data, but an expression, unique to each field, of what counts as a disappointing result.

## Skewing the record

The reluctance of scientists to share negative results, and the twinned reluctance of journals to publish them, arise from biases that we can and should confront. This does not mean we should not be discerning. As BioRxiv and MedRxiv co-founder Richard Sever has observed, "data are not a democracy." Some results are just flimsy and should not be included in the scientific literature.

Life Science Alliance LLC, New York, NY, USA

Correspondence: t.fessenden@life-science-alliance.org

Furthermore, some findings are simply more compelling and hold the attention of researchers, journal editors, peer reviewers, and readers more than others. The qualitative and subjective evaluation of research results are not something to be eliminated at all times and in all forms. Yet the tendency to discount null results demands self-reflection.

The detrimental impacts of the bias against negative data are plain. Research dead ends, flawed reagents, and faulty hypotheses all persist and consume resources for longer than they should. By deciding, or being informed by a journal, that negative results are not sufficiently interesting to publish, researchers also lose recognition for their efforts and potential feedback. In a large and detailed survey, respondents stated that reading null results spurred their own thinking and generated new hypotheses (Springer Nature, 2025). Interested readers from different fields may find value in a publication that is entirely unanticipated by the authors (or for that matter by journal editors). Why would null results be an exception? Authors can't know in advance how their work will impact the scientific community, yet there is a stubborn assumption that only positive results can be of practical use or broad interest.

There is no single culprit promoting this assumption and keeping important negative results out of the scientific literature. Instead, this trend is supported through an insidious collaboration of incentives and assumptions. Journals assume readers are less interested in null results—that they will not be read or cited. Meanwhile, authors assume that journals view negative results unfavorably (Springer Nature, 2025). Whether or not this assumption is merited, many journals do not explicitly welcome such manuscripts. This is not to scold either party for considering the level of reader interest in a manuscript. But the prevailing view that readers could not possibly find null results interesting is objectively false. Further, some negative results provide an essential correction to the scientific record in form of refutations. As long as authors and journals mutually reinforce the perception that the other does not welcome negative results, those studies will not find their way easily into the scientific record.

## Negative results at *Life Science Alliance*

This month, *LSA* presents a collection of articles that include or explicitly focus on null results. Some of these serve to refute claims from prior papers, for instance on cytosolic DNA sensing by cGAS/STING, or on chloride channels and mucus production. I warmly congratulate these authors for composing and submitting these unique papers. I hope readers will enjoy them and perhaps find unexpected value in the observations they convey.

## Author Contributions

T Fessenden: conceptualization and writing—original draft, review, and editing.

## Conflict of Interest Statement

The author declares no conflicts of interest.

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
