## [Reviewer comments · Life Science Alliance]

Reframing "Disappointing" Data at Life Science Alliance

Author information redacted

DOI: <https://doi.org/10.26508/lsa.202603674>

Corresponding author(s): Author information redacted

Review Timeline:	Submission Date:	2026-02-18
	Accepted:	2026-02-18

Scientific Editor: Sarita Hebbar

Transaction Report:

February 18, 2026

RE: Life Science Alliance Manuscript #LSA-2026-03674

Tim Fessenden
Life Science Alliance LLC
The Rockefeller University Press
New York, New York

Dear Dr. Fessenden,

Thank you for submitting your Editorial entitled "Reframing "Disappointing" Data at Life Science Alliance". It is a pleasure to let you know that your manuscript is now accepted for publication in Life Science Alliance. Congratulations on this interesting work.

DISTRIBUTION OF MATERIALS:

Again, congratulations on a very nice paper. I hope you found the review process to be constructive and are pleased with how the manuscript was handled editorially. We look forward to future exciting submissions from your lab.

Sincerely,

Sarita Hebbar, PhD
Scientific Editor
Life Science Alliance
<http://www.lsjournal.org>

February 18, 2026

RE: Life Science Alliance Manuscript #LSA-2026-03674

Tim Fessenden
Life Science Alliance LLC
The Rockefeller University Press
New York, New York

Dear Dr. Fessenden,

Thank you for submitting your Editorial entitled "Reframing "Disappointing" Data at Life Science Alliance". It is a pleasure to let you know that your manuscript is now accepted for publication in Life Science Alliance. Congratulations on this interesting work.

DISTRIBUTION OF MATERIALS:

Again, congratulations on a very nice paper. I hope you found the review process to be constructive and are pleased with how the manuscript was handled editorially. We look forward to future exciting submissions from your lab.

Sincerely,

Sarita Hebbar, PhD
Scientific Editor
Life Science Alliance
<http://www.lsjournal.org>